# Renal and Endocrine Responses to Arm Exercise in Persons with Cervical Spinal Cord Injury

**DOI:** 10.3390/jcm12041670

**Published:** 2023-02-20

**Authors:** Yuki Mukai, Takashi Kawasaki, Yoshi-ichiro Kamijo, Kazunari Furusawa, Tatsuru Ibusuki, Yuta Sakurai, Yukihide Nishimura, Yasunori Umemoto, Fumihiro Tajima

**Affiliations:** 1Department of Rehabilitation Medicine, School of Medicine, Wakayama Medical University, Wakayama 641-8509, Japan; 2Department of Rehabilitation Medicine, Graduate School of Medical Science, Kyoto Prefectual University of Medicine, 465 Kajii-cho, Kawaramachi-Hirokoji, Kamigyo-ku, Kyoto 602-8566, Japan; 3Department of Rehabilitation Medicine, Dokkyo Medical University Saitama Medical Center, Saitama 343-8555, Japan; 4Department of Rehabilitation Medicine, Kibikogen Rehabilitation Center for Employment Injuries, Kibichuo-cho, Okayama 716-1241, Japan; 5Research Center of Sports Medicine and Balneology, Nachikatsuura Balneologic Town Hospital, Tenma-Nachikatuura-cho, Wakayama 649-5331, Japan

**Keywords:** ADH, glomerular filtration rate, adrenaline, sympathetic nerves

## Abstract

The aim of this study was to assess renal functions and endocrine responses to arm exercise in persons with cervical spinal cord injury (CSCI) under euhydrated conditions (free drinking of water), and to determine the physiological effects of exercise on renal function in these subjects. Eleven CSCI individuals (spinal lesions between C6 and C8, American Spinal Injury Association impairment scale A) and nine able-bodied (AB) persons rested for 30 min before performing 30 min arm-crank ergometer exercises at 50% of their maximum oxygen consumption, followed by 60-min of rest/recovery. Urine and blood samples were collected before and immediately after the exercise and recovery period. The CSCI patients showed no increase in plasma adrenaline and plasma renin activity compared with the AB controls, but showed similar changes in plasma aldosterone and the plasma antidiuretic hormone in response to the exercise. Creatinine clearance, osmolal clearance, free water clearance, and the fractional excretion of Na^+^ did not change during exercise in both groups of subjects, however free water clearance in the CSCI group was higher than in the AB group throughout the study. These findings suggested that activated plasma aldosterone without an increase in adrenaline or renin activity during exercise in CSCI individuals may reflect an adaptation to the disturbance of the sympathetic nervous system to compensate for renal function. As a result, no adverse effects of exercise on renal function in CSCI patients were observed.

## 1. Introduction

Exercise in persons with cervical spinal cord injury (CSCI) and thoracic and lumbar spinal cord injury (SCI) should improve health outcomes as well as wellbeing and social activities. In terms of individuals with SCI, 2 persons who continued strenuous wheelchair sports activities showed increased VO_2_max by 43% and 45% after 20 years [1]. The health benefits of sport participation by athletes with disabilities have been well recognized [2]. Compared with SCI, CSCI is associated with the severe impairment of efferent sympathetic nerve activity, and often causes various physiological problems both at rest and during exercise. The management and care of individuals with SCI includes bladder management and the prevention of renal dysfunction. Urinary tract infections are one of the main causes of death for individuals with SCI [3]. Exercise in such individuals could elicit an increase in circulating catecholamines and the constriction of renal arteries, potentially resulting in a decrease in the glomerular filtration rate (GFR). Based on this background, the effect of exercise on renal function has been the focus of attention in SCI. Previous studies have reported the tendency for a significant, though brief, decrease in GFR in healthy able-bodied subjects (AB) at a high work intensity [4], with a 9–15% decrease in GFR when AB individuals exercise to exhaustion [5], compared with no change in GFR in SCI patients during a 2 h arm ergometry exercise at 60% of their maximum oxygen consumption [6].

Since renal sympathetic nerve activity contributes to the regulation of Na^+^ reabsorption in the distal renal tubules [7,8], we were concerned about the effect of exercise on renal function in CSCI individuals. The purpose of this study was to assess the effect of exercise on renal function in individuals with CSCI.

## 2. Materials and Methods

### 2.1. Research Design

This was an observational study.

### 2.2. Subjects

The study subjects were 11 persons with CSCI and 9 AB individuals. Physical and neurological examinations confirmed a medically healthy state in each subject except for CSCI-related physical problems (e.g., paralysis with leg edema, neurogenic bladder, constipation) in the CSCI group. Women were excluded to avoid the possible influence of female cyclical hormonal changes on various body organs. The affected areas in CSCI subjects included C6 to C8, with scale A impairment according to the American Spinal Injury Association (i.e., tetraplegia due to complete cervical spinal cord injury). None of the 20 subjects was on any medication that could influence the study results.

Table 1 summarizes the characteristics of the two groups. There were no differences between the CSCI and AB subjects with respect to age, height, body weight, and body surface area. The study protocol was approved by the Human Research Committee of Wakayama Medical University School of Medicine, and each subject provided a signed informed consent form before participating in the study.

### 2.3. Study Protocol

To measure the maximum oxygen consumption (VO_2_max), each subject underwent an arm-crank ergometer exercise test in advance using the multistage load method. To allow them to perform the exercise, we placed the subjects’ hands on the ergometer handles, then wrapped the two together with a bandage, i.e., fixed the hands to the handle. This allowed the tetraplegic subjects to perform the exercise using the biceps and deltoid or other arm muscles, even when the triceps and flexor digitorum profundus muscles were paralyzed. All the subjects were instructed to abstain from alcohol consumption and intensive exercise for at least 24 h before the study, to have their normal dinner before 10:00 pm on the day before the exercise, and to avoid food thereafter until the end of the study. The subjects reported to the laboratory at 8:00 am and were set up for the ECG recording; they were asked at 9:00 am to urinate, and then rested in a quiet room (room temperature 26 °C) for 30 min. This was followed by the collection of blood and urine samples. The subjects then performed the arm-crank ergometer exercise (818E, Hand Ergometer, Monark, Sweden) for 15 min at 50% VO_2_max, rested for 2 min to prevent exhaustion, and then completed another 15 min exercise (total exercise duration: 30 min). Immediately after the end of the exercise, venous blood and urine samples were collected and the subjects were allowed to recover in a seated position for 60 min. This was followed by the collection of recovery venous blood and urine samples. The time duration from before the start of the exercise to immediately after the exercise was designated as P1, while the time from immediately after the exercise to the end of the following 60 min was termed P2.

The consumption of food was not allowed throughout the study. Urine samples were provided by the AB subjects at preselected time points and from urinary catheters or spontaneous voiding in the CSCI subjects. Each subject was also instructed to record their volume of water intake throughout the test period. The latter and the volume of voided urine were calculated, and the time periods among the different time points were also calculated. Furthermore, the levels of creatinine, osmolality, and Na^+^ were measured in the collected urine samples.

Blood samples were collected immediately before urine sampling, using standard procedures. Each sample was divided into two portions: 8 mL placed into a chilled Na2 EDTA-containing vacutainer for the analysis of plasma adrenaline, plasma renin activity (PRA), plasma aldosterone (Pald), plasma antidiuretic hormone (ADH) and a second portion of 5 mL collected into lithium heparin-treated syringes for the estimation of creatinine, plasma osmolality, and Na^+^. 

### 2.4. Blood and Urine Sample Analysis

Immediately after collection, each sample was centrifuged at 4 °C and placed on ice and stored at −80 °C until the analysis. The urine samples were also immediately stored at −80 °C until the analysis. All the measurements were conducted within 30 days of collection. 

After extraction from plasma with alumina, plasma adrenaline was measured by high-performance liquid chromatography (HPLC system JASCO Corporation, Tokyo, Japan) using the procedure described previously by Hunter et al. [9] with minor modifications. A freezing point depression osmometer (model AUTO&STAT OM-6030, Arkray, Kyoto, Japan) was used to measure the plasma and urine osmolalities. The ion selective electrode method was applied to measure the plasma and urine Na^+^ levels, using an autoanalyzer (BM8060 JEOL Ltd, Tokyo, Japan)Furthermore, the creatininase-creatinase-sarcosine oxidase-POD method was applied for the measurement of plasma and urine creatinine levels, using the above autoanalyzer. The PRA was measured using a method based on the generation of angiotensin I in plasma samples over 60 min at 37 °C followed by the measurement of angiotensin I by a double-antibody ^125^I-radioimmunoassay using a gamma counter instrument (ARC 950,HITACHI Ltd, Tokyo, Japan). A double-antibody ^125^I-radioimmunoassay was applied to measure the plasma ADH using the gamma counter instrument (ARC 950 HITACHI Ltd, Tokyo, Japan), while a competitive solid-phase ^125^I-radioimmunoassay technique was applied to measure Pald, using the kit(SPAC-S Aldosterone kit, TFB inc, Tokyo, Japan).

The following equation was used to calculate the creatinine clearance: (C_Cr_ mL/min): C_Cr_ = UCr × V/PCr (UCr: urine creatinine level; PCr: plasma creatinine level; V: urine flow volume). The urinary osmolar excretion was estimated in UosmV mOsm/min, using the formula: UosmV = Uosm × V (Uosm: urine osmolality; V: urine flow volume). The osmolal clearance (in Cosm mL/min) was computed using the equation: Cosm = UosmV / Posm (UosmV: urinary osmolar excretion; Posm: plasma osmolality).

The following formula was used to estimate the free water clearance (in CH2O mL/min): CH2O = V-Cosm (V: urine flow volume; Cosm: osmolal clearance). Finally, we used the following formula to determine the fractional excretion of Na^+^ (FE_Na_%): FE_Na_ = [UNa^+^ × PCr]/[PNa^+^ × UCr] × 100 (PNa^+^: plasma Na^+^; UNa^+^: urinary Na^+^; PCr: plasma creatinine; UCr: urinary creatinine level).

### 2.5. Statistical Analysis

The data were expressed as the mean ± standard error of the mean (SEM). The differences were assessed using a 2 × 4 repeated measures analysis of variance (ANOVA) followed by the Tukey–Kramer test for the parameters that showed significant differences (*p* < 0.05) between the pre-exercise and each time period, and between the CSCI and AB groups. A *p* value less than 0.05 denoted the presence of a significant difference. All the statistical analyses were conducted using the SPSS (Statistical Package for Social Sciences) software for Windows (version 23, IBM SPSS, Chicago, IL, America). 

## 3. Results

### 3.1. Effects of Exercise on Plasma Osmolality and Na^+^

In the AB group, plasma osmolality and Na^+^ were both significantly lower 60 min after the 30-min exercise, relative to the baseline (Figure 1). Plasma osmolality in the CSCI group did not change throughout the study whereas plasma Na^+^ was significantly lower 60 min after the 30-min exercise, compared with the pre-exercise control value (Figure 1). 

### 3.2. Endocrine Responses

The plasma adrenaline level was significantly higher at the end of the 30-min exercise in the AB subjects but recovered to the baseline level 60 min after the exercise (Figure 2a). On the other hand, no changes were seen in the level of this hormone throughout the study in the CSCI subjects (Figure 2a). A similar trend was noted in plasma renin activity in both the AB and CSCI group (Figure 2b). On the other hand, the plasma aldosterone levels were significantly higher immediately after the exercise in both the AB and CSCI subjects, but returned to the baseline levels 60 min after exercise, in both groups of subjects (Figure 2c). A similar trend was noted in the plasma ADH levels in both the AB and CSCI subjects (Figure 2d), though the plasma level of this hormone was significantly lower in the CSCI group than in the AB group at each time point throughout the study (Figure 2d).

### 3.3. Renal Response to Exercise

While changes in water intake, urine flow, total urinary sodium discharge, and creatinine clearance were noted between the P1 (the time from before the start of exercise to immediately after exercise) and the P2 (the time from immediately after exercise to the end of recovery) periods in both the CSCI and AB subjects, these changes were not statistically significant (Figure 3). Furthermore, the Cosm levels during exercise (P1) were significantly lower than those recorded during the recovery period (P2) in both groups (Figure 4a). The levels of CH2O did not change in both groups throughout the study; however, they were significantly higher in the CSCI group than in the AB group in both phases of the study (P1 and P2, Figure 4b). 

## 4. Discussion

Our study was performed to determine the effects of CSCI on renal and endocrine responses to arm exercise. Previous studies have examined renal function during leg and arm exercises in persons with SCI and AB individuals [6]; however, there are no reports on the effects of exercise on the endocrine and renal functions in individuals with CSCI. The major new findings of our study were: (1) CSCI has no effect on the renal response to arm exercise; (2) CH2O was higher in individuals with CSCI than in AB individuals both at rest, during exercise, and post exercise; (3) CSCI blunts the response of plasma adrenalin and renin activities to arm exercise, relative to that seen in AB individuals; (4) CSCI has no effect on the exercise-induced response in terms of plasma aldosterone levels; and (5) while ADH was lower in the CSCI group than in the AB group throughout the study, the response of this hormone to exercise was not affected by CSCI. 

Plasma adrenalin did not increase during the arm exercise in the CSCI group compared with the increase seen in the AB group. This finding might be due to the inhibition of renal sympathetic nerve activity in CSCI individuals [10]. Renal sympathetic nerve activity contributes to the regulation of Na^+^ reabsorption in the renal distal tubules. It is likely that the lack of renal sympathetic innervation in individuals with CSCI is the major physiopathological mechanism of the differences in the renal functional response between AB and CSCI individuals observed in this study. Previous studies have reported that the identical response of adrenalin during arm exercise in individuals with CSCI did not increase the plasma glucose concentration and resulted in a decrease in the serum insulin concentration [11]. Insulin is known to regulate the transport functions in the renal proximal tubule [12]. However, the present results suggested that the decrease in insulin level did not affect renal function in individuals with CSCI. 

In the AB subjects, exercise sets a dominant sympathetic systemic condition together with a fall in renal blood flow [13] and hormonal changes that decrease urine flow. Leg exercise in AB subjects under non-hydration conditions induces a rise in plasma osmolality with a resultant identical rise in the levels of CH2O and Cosm [4,14,15,16], due to water loss through expiration and sweating. However, all the subjects in the present study were allowed to drink water ad libitum during the arm exercise and, accordingly, there were no significant changes in urine flow or plasma osmolality. Thus, CH2O also did not change significantly during arm exercise.

In the present study, the renal functional response to exercise in the CSCI group was almost identical to that of the AB group, with the exception that CH2O was higher in the CSCI group than in the AB group throughout the experiment. The renal response to exercise in the present study was like that reported earlier by Tajima et al. [17], that CH2O tended to be higher in CSCI than AB individuals. In the present study, ADH increased during exercise in both groups, however, it was significantly lower in the CSCI group than in the AB group. Szollar et al. [18] reported that CSCI individuals lack the normal diurnal variation in ADH and appear to have generally depressed ADH levels compared to AB individuals. The lower ADH in CSCI individuals suggests a tendency toward diuresis and hypotonic urine, leading to higher levels of CH2O. It is not clear why the ADH level is lower in CSCI individuals than in AB individuals; however, it may reflect the adaptation of CSCI individuals to maintain electrolyte homeostasis by keeping ADH secretion in the blood at a low level in advance to prevent hypernatremia due to dehydration.

In the present study, no significant changes were noted in plasma osmolality during exercise in the CSCI group; however, the ADH level was significantly higher in the CSCI group. ADH secretion is stimulated by a decrease in effective blood volume or an increase in extracellular fluid osmolarity [19]. It is likely that exercise induces an increase in cardiac output in CSCI individuals, with an incomplete return of the increased cardiac output to the atrium as a result of the transected efferent sympathetic nerves at the cervical spinal lesion and the absence of peripheral vessel contraction in CSCI. Such a change can result in a decrease in effective blood volume with a consequent decrease in ADH during exercise in CSCI individuals. Our finding on ADH is in agreement with a previous study that showed an increase in ADH in AB individuals during highly intensive exercise [4]. 

In the present study, the AB subjects showed an increase in renin activity, similar to adrenaline, during arm exercise, but no changes were noted in both hormones in the CSCI subjects. This was in contrast to the observed exercise-induced increase in plasma aldosterone levels in both groups of subjects. The sympathetic nervous system affects renin secretion via the renal nerves and through the actions of circulating catecholamines. In another study, Kawasaki et al. [6] reported that a 2 h arm crank ergometer exercise increased plasma aldosterone and lowered PRA activation during exercise in SCI individuals, but not in AB individuals. They suggested that the activated aldosterone response during exercise in SCI subjects was probably the result of an adaptation of the sympathetic nervous system to stress to maintain normal renal function. Thus, the enhanced response of Pald in CSCI individuals could reflect an adaptation of the disordered sympathetic nervous system to maintain renal function, similar to in SCI individuals [5]. In this regard, dopamine inhibits angiotensin II-stimulated aldosterone secretion in normal human subjects [20]. Our results suggest a lack of suppression of aldosterone secretion in CSCI individuals in the absence of an increase in adrenaline, with a resultant increase in plasma aldosterone levels, similar to that seen in AB levels. 

### Study Limitation

Based on ethical considerations, the participating CSCI subjects were allowed free access to water to avoid possible complications. This aspect of the study design may limit the full interpretation of the data, because water intake prevented variations in plasma osmolality, urine flow, and Na^+^ excretion. Thus, the design of the present study was different from those reported previously in terms of the renal response to exercise in AB individuals conducted under water restriction conditions.

## 5. Conclusions

We have demonstrated in the present study that arm-crank ergometer exercise for 30 min did not adversely affect renal function in CSCI individuals. Plasma aldosterone increased in the CSCI subjects without an increase in adrenaline or renin activity. The response of the plasma aldosterone in the CSCI subjects could reflect an adaptation of the disordered sympathetic nervous system to maintain renal function.

## Figures and Tables

**Figure 1 jcm-12-01670-f001:**
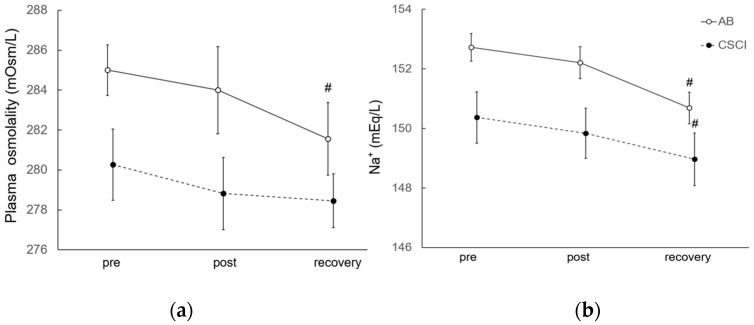
Plasma osmolality and Na^+^ concentrations before (pre), immediately after (post), and 60 min after (rec) exercise. Data are mean ± SEM. # *p* < 0.05 vs. the pre value of the same subject group. AB: able bodied subjects, CSCI: cervical spinal cord injury. (**a**) Plasma osmolality; (**b**) Na^+^.

**Figure 2 jcm-12-01670-f002:**
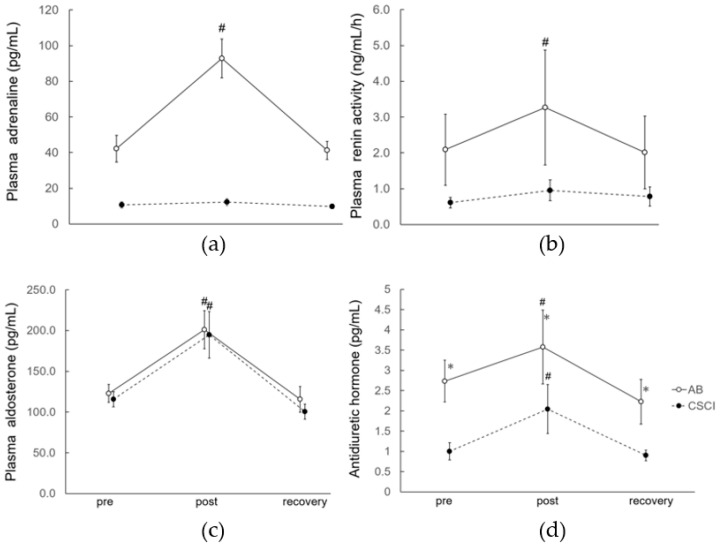
Plasma adrenaline, renin activity, aldosterone, and antidiuretic hormone before (pre), immediately after (post), and 60 min after (rec) exercise. Data are mean ± SEM. # *p* < 0.05 vs. pre value of the same subject group, * *p* < 0.05 vs. the value of CSCI at the same time period. Abbreviations as in Figure 1. (**a**) Plasma adrenaline level; (**b**) plasma renin activity; (**c**) plasma aldosterone level; and (**d**) plasma ADH.

**Figure 3 jcm-12-01670-f003:**
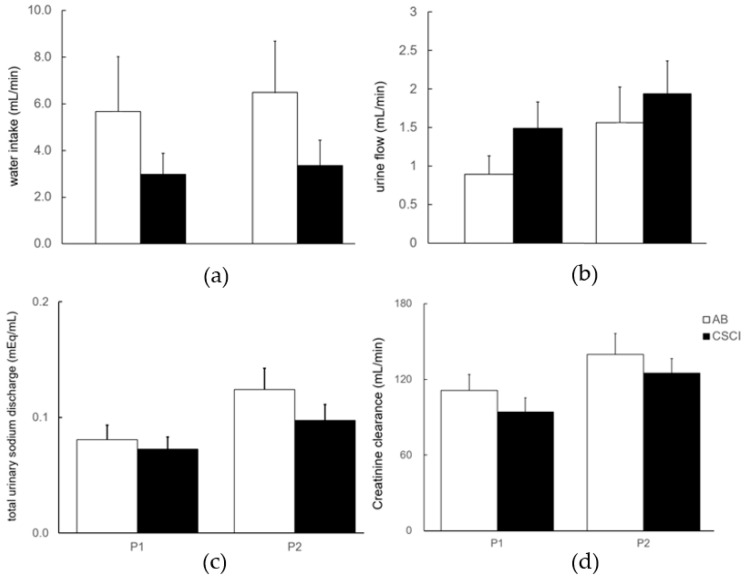
Water intake volume, urine flow, total urinary sodium discharge, and creatinine clearance at the two different time periods. Note the lack of statistically significant changes in the four parameters between the P1 and P2 periods. P1: time period from before the start of exercise to immediately after exercise, P2: time period from immediately after exercise to the end of the 60-min recovery period. Data are mean ± SEM. (**a**) shows water intake, (**b**) shows urine flow, (**c**) shows total urinary sodium, and (**d**) shows creatinine clearance.

**Figure 4 jcm-12-01670-f004:**
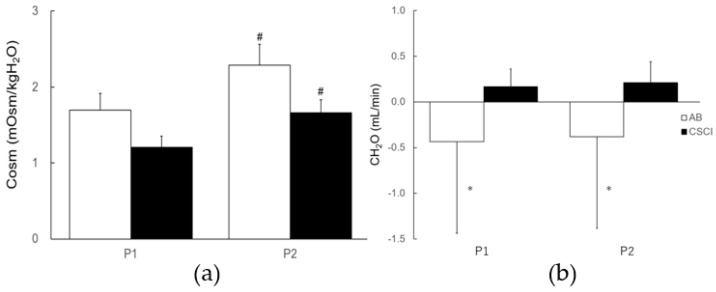
Cosm and CH2O at the two different time periods. Data are mean ± SEM. # *p* < 0.05 vs. P1, * *p* < 0.05 vs. CSCI. Abbreviations as in Figure 1 and Figure 3. Cosm levels are shown in (**a**), and the levels of C_H2O_ are shown in (**b**).

**Table 1 jcm-12-01670-t001:** Characteristics of subjects.

	n	Age, Years	Height, cm	BW, kg	BSA, m^2^
AB	9	42 ± 4	171 ± 2	68.5 ± 4.0	1.8 ± 0.1
CSCI	11	43 ± 4	173 ± 2	61.2 ± 4.6	1.7 ± 0.1

Data are mean ± SD. BW: body weight. BSA: body surface area.

## Data Availability

The datasets generated during the current study are available from the corresponding author upon reasonable request.

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
