# Peer review of "Renal and Endocrine Responses to Arm Exercise in Persons with Cervical Spinal Cord Injury"

_jcm, 2023, doi:10.3390/jcm12041670_

Round 1

Reviewer 1 Report

This article examines the effects on the kidney and endocrine system of patients with cervical spinal cord injury (CSCI) by performing arm exercise. As a result, the serum aldosterone level was equivalent to that of healthy able-bodied subjects (AB), but the adrenaline and renin activity levels were not. It was concluded that this may be an adaptive response in CSCI patients as a result of a compromised sympathetic nervous system.

I believe that your paper clearly advances the argument in terms of experimental methods, interpretation of the results, and discussion.

I consider your article to be a highly complete one.

Reviewer 3 Report

The authors propose their study on the effect the exercise might have on the renal and endocrine functions in patients with cervical spinal cord injury (CSCI).

Introduction provides sufficient information on the topic, but lacks references within the first two thirds. This should be improved. Also, title should not be repeated as the first sentence in Abstract.

The level of injury is usually described by the vertebral column affection, rather then the roots (C6-8). Also, tetraplegic is used instead of complete injury in the explanation of the ASIA). C6 and below injuries result in paraplegia if not recovered, but it is also important to note that some extent of hand function impairment might be present, thus compromising the use of ergometer. This should be clarified as whether the CSCI subjects had this impairment or not, otherwise it should be a limitation.

Minor English language correction and more attention is advised. The title lacks the word responses.
